# Impact of Cruising for Parking on Travel Time of Traffic Flow

**Yating Zhu [1] , Xiaofei Ye [1],* , Jun Chen [2] , Xingchen Yan [3] and Tao Wang [4]**

[1] Ningbo Collaborative Innovation Center for Port Trade Cooperation and Development, School of Maritime and Transportation, Ningbo University, Fenghua Road 818#, Ningbo 315211, China; yatingzhu9422@gmail.com

[2] School of Transportation, Southeast University, Si Pai Lou 2#, Nanjing 210096, China; chenjun@seu.edu.cn

[3] College of Automobile and Traffic Engineering, Nanjing Forestry University, Nanjing 210037, China; xingchenyan.acad@gmail.com

[4] School of Architecture and Transportation, Guilin University of Electronic Technology, Nanjing 541004, China; wangtao@sue.edu.cn

\* Correspondence: yexiaofei@nbu.edu.cn; Tel.: +86-1526-785-9815

**Abstract:** Cruising for parking creates a moving queue of cars that are waiting for vacated parking spaces, but no one can see how many cruisers are in the queue because they are mixed in with normal cars that are actually going somewhere. In order to mitigate the influence of cruising for parking on the normal cars, the park-and-visit cruising tests with GPS and cameras was applied to collect the behavior of the cruisers, and the videotapes of traffic flows were used to measure the volume of cruising cars and the traffic status of normal cars, simultaneously. On this basis, a parking time model based on proportional hazard-based duration model was proposed, and the factors affecting cruise for parking were analyzed, including the volume, search time, speed, acceleration, lane-change frequency, and distracted time of the cruising car. The multiple linear regression model was also established to compare with proportional hazard-based duration model results. The results indicated that between 9 and 56 percent of the traffic was cruising for parking, and the average search time was about 6.03 min. The low-speed, volume, high acceleration frequency, and lane-change times of cruising cars have a negative effect on shortening travel time of the normal traffic flow. Conversely, high-speed of cruising cars has a positive effect on shortening travel time of traffic flow. Moreover, travel time changes in varying degrees due to various factors. Under postulated conditions, the model can be used to estimate the travel time. It is hoped that this study will contribute to improve the planning and management of cruising for parking.

**Keywords:** parking management; cruising for parking; Cox proportional hazard-based duration model; travel time

## 1. Introduction

With the increase of motor vehicles in the central business district (CBD) of the city, more and more cars would not only bring traffic congestion, but also parking problems. If all the parking spaces are occupied, drivers must cruise to find a space vacated by a departing car. Cruising for parking probably began soon after the vehicles were invented. Actually, a mobile queue of cars is generated by cruising for parking spaces. They usually keep a low speed and frequently change lanes and acceleration. The cruisers are mixed in with other cars that are actually going somewhere. Because cruising is invisible and no one can see how many cars are cruising in the queue, traffic congestion and pollution in CBD of the city are deteriorating by the cruisers. Nevertheless, a few researchers thought that cruising for parking could not be neglected as a source of traffic congestion, and the volume and

the time of cruising for parking should be estimated. Notably, the problem of "parking difficulty" in the city CBD is extremely prominent, forcing the arriving vehicles to frequently exhibit low-speed cruising behavior, which is mixed into the normal traffic flow. These cruising behaviors aggravate traffic congestion and increase traffic pollution. This issue has attracted wide attention from scholars. For example, Donald found that about 30% of vehicles in the traffic flow were cruising for parking, and the average time to find a curb space was about 8 min. Every year in Los Angeles, the whole city wasted nearly 1.61 million kilometers of low-speed vehicles, wasted 95,000 h, consumed 47,000 gallons of fuel, and emits created 730 tons of $CO_2$ emissions [1]. Obviously, cruising for parking congests traffic, pollutes the air, and creates $CO_2$ emissions.

Because a moving queue of cars is created by cruising and are mixed in with other normal cars that are actually going somewhere, the influence of cruising cars on the normal traffic flow is hidden behind. In order to mitigate the influence of cruising cars, its effect must first be measured reliably. Therefore, the cruising in park and visit tests of GPS trajectories were applied to collect the behavior of the cruisers, and the videotape of traffic flows was used to measure the volume of cruising cars and the traffic status of normal cars, simultaneously. Using a sample of 450 observations during peak and off-peak hours on two weeks, we estimated that between 9 and 56 percent of the traffic was cruising for parking, and the average search time was about 6.03 min. Then, a proportional hazard-based duration model is proposed to analyze the influential factors related to cruising for parking. The findings could explain how cruising cars add to the normal traffic that is already congested, and quantifies the worse situation of the influence. Cruising cars increased traffic congestion by reducing speed, increasing number of accelerations, increasing number of lane-changes, and increasing the distracted time of cruising drivers.

The contributions of this study: (1) Showed the impact of cruising vehicles on normal traffic flow. (2) Explained how cruise cars increase traffic congestion.

At the same time, it will be convenient to lay a foundation for more microscopic research on cruising for parking behavior in the future, and it can provide some theoretical foundation and improvement directions for future traffic congestion and parking problems.

## 2. Literature Review

The International Handbook for On-Street Parking Management mentions that when the parking is close to saturation (the occupancy rate is above 85%), it is difficult for the driver to find the parking space for parking, thereby increasing the traffic burden of the congested area [2]. Parking traffic in the saturated congestion zone of parking is usually higher than 30%. In Richard Arnott, Ommeren JNV, parking cruising behavior is considered to increase traffic congestion [3,4].

In a study of the impact of parking rate on parking cruising behavior, Ommeren JNV first used an identification methodology based on house prices for Amsterdam to empirically test residents' willingness to pay for cruising fees [5]. Martijn B.W. Kobus established a parking charge model to reduce parking cruising behavior based on the price elasticity of street parking demand, parking duration, walking distance, and other factors, and proposed measures to guide short parking in the road and stop when the road is busy [6]. When studying the relationship between parking cruising behavior and parking price, the influencing factors of parking price should also be considered. Wang Xin et al. used the survey data and structural equation modeling method to establish the structural equation model of the parking price influencing factors from the parking lot, traffic environment, government, and market. The results show that the parking demand and parking difficulty directly affected the parking price, while the parking management intensity, the completeness of parking policies and regulations, the price tolerance of residents, the length of parking, the congestion on roads, and the level of public transportation indirectly affect the parking price [7]. Mei Zhenyu et al. constructed an in-road parking pricing model based on parking choice behavior and applied the model to optimize the utilization rate of static traffic facilities in the parking planning of Tongling city [8]. Zhen (Sean) Qian used Linear Programming method to get the System Optimal (SO) parking flow

pattern and SO parking price of the system. The results showed that the optimal pricing essentially balanced parking congestion (i.e., cruising time) and convenience [9].

In a study of parking cruising, behavior is generated by people. From the perspective of drivers, Russel G. Thompson et al. proposed a parking lot selection model based on drivers' parking cruising behavior. The application results show that drivers' rich experience does not reduce parking cruising behavior [10]. Ding Huan et al. built a parking cruising behavior model on the road considering the influence of time value, and the results showed that the search time of non-work trips was longer than that of work trips [11]. Arnott analyzed the change of parking search time from namely walking distance and parking space supply and demand, and established a model with parking space occupancy rate as the optimal target. According to the research, activities in a specific period of time will aggravate the generation of parking cruising behavior [12]. Yan Hai and Yang Xiaoguang proposed a time-benefit model of parking choice behavior to solve the theoretical problems of parking prediction and management in special activities [13]. Arnott et al. used the simple cellular automata condition simulation modeling to study that the higher the road occupancy rate, the longer the actual cruising time [14]. Mannini used FCD data of the detection vehicle to identify the cruising vehicles, modeling their cruising time. This model is suitable for real-time support of user information and dynamic routing, and also applies to the off-line assessment transmission plan [15].

In study of the travel time model. Xiang Yan calibrated a joint model of travel mode and parking location choice, and investigated the main policy variables including parking cost, parking search time, and exit time. The results showed that the combination of price and policy measures to reduce search and evacuation time was better than the single implementation [16]. N. Geroliminis proposed a cruising for parking model. This model considered the dynamics of different types of vehicles when driving or cruising to the destination. The results showed that cruising for parking affected all vehicles in the road network [17].

In a study of the impact of the driving path of parking cruising to and from the parking on traffic, Webster used simulation software to simulate the impact of on-street parking on straight traffic, and concluded that the turnover rate of on-street parking space affected traffic flow delays [18,19]. Saeed et al. applied logit model to analyze the driving path of traffic to and from the parking lot, and obtained data that indicated that parking cruising behavior would increase traffic pressure [20].

Previous research mainly focused on the parking cost, but ignored the variables unrelated to price, such as parking search time and cruising for parking behavior. In this paper, the cruising for parking behavior was studied. The relationship between the travel time and cruising behavior was analyzed with the field data.

## 3. Data Collection

Park-and-visit tests with GPS and cameras were applied to capture the behavior of cruising for parking in this study. According to the parking lot survey, 16:00–19:00 on weekends and holidays in the evening had a high probability of parking cruising behavior. Therefore, the survey time selected both weekends and holidays. The data used were collected in the Tianyi Square CBD of Ningbo city during the peak and off-peak hours (16:00–19:00). As shown in Figure 1, the experimental area of Tianyi Square is surrounded by four roads, Jiangxia Street, Zhongshan Road, Kaiming Street, and Dashani Street. There are three parking lots for an experimental vehicle to choose in the study area. The nearest parking lot to Tianyi Square is the first choice of an experimental vehicle. If all the parking lots at the first choice were occupied, the experimental vehicle starts to cruise to find a vacated parking space around the study area. The following methods are applied to data collection.

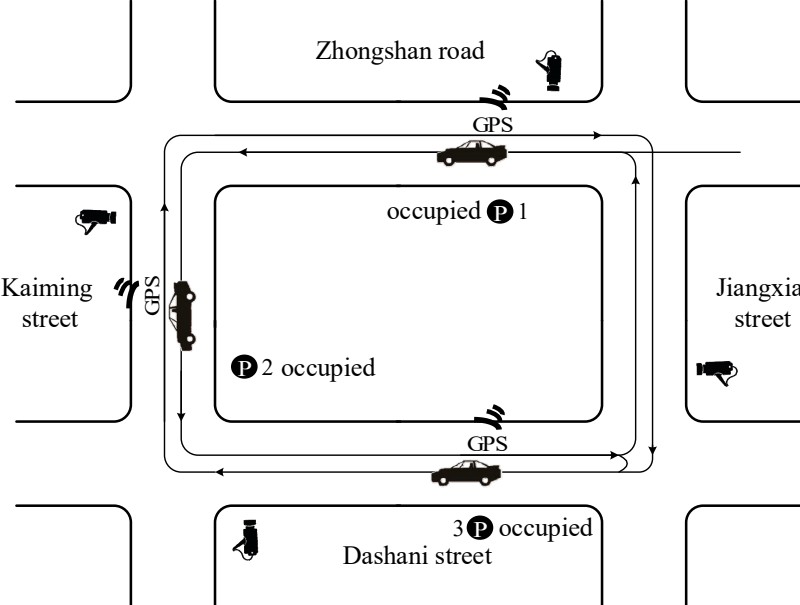

**Figure 1.** Investigation Plan.

(1) In order to measure the behavior of the cruising car, the experimental vehicle was equipped with GPS and two cameras [21]. The experimental vehicle first drove into the nearest parking lot to the destination. If all the parking spaces at the first choice were occupied, the experimental vehicle starts to cruise to find a vacated parking space around the study area. Until the experimental vehicle finds the available parking space, the park and visit test could be finished once and drive out of the study area to start a new test. GPS trajectories were applied to calculate the speed, acceleration, and lane-change times of cruising car. The cruising or search time could also be collected by the experimental car for every park and visit test. One camera was used to record the distracted time of the driver, which was paying more attention to searching for vacated parking lot, and not focusing on the driving behavior. The other camera was recording the traffic flow during the cruising process. The 456 tests were collected during peak hours in 14 days.

(2) In order to measure the volume of cruising car and the traffic status of normal cars, the cameras were set up at the observation points as showed in Figure 1. If cars repeatedly passed observation points at two locations in the study area or appeared in the camera of experimental vehicle at the same parking lot, they were counted as the cruising cars. Moreover, travel time and volume of normal traffic mixed with the cruising cars could be calculated by the videotape. The experimental vehicle and normal traffic flow were matched simultaneously in each episode. The data of each episode were conducted.

### 3.1. Survey Results from Park-and-Visit Tests

According to the park-and-visit tests, the average search time of vehicles was 6.03 min, the average speed of cruising car in the road network was 13.53 km/h, the average acceleration was 0.25 m/s$^2$, the average acceleration times was 27.41 times, the average lane-change times was 4.79 times, and the average distracted time was 3.53 s. According to the variance analysis, cruising cars had a high frequency of speed-change and lane-change behavior. More characteristic parameters of cruising cars, such as cruising track, parking cruising duration, and lane change times, were obtained in Table 1.

**Table 1.** Characteristics of Test Sections.

| Test Section | Minimum | Maximum | Mean | Variance | Remarks |
|---|---|---|---|---|---|
| Search time (min) | 1.350 | 29.710 | 6.030 | 3.859 | Total track length. Search time of cruising cars is generated by looking for parking space. |
| Speed of cruising car (km/h) | 0.100 | 52.130 | 13.530 | 57.541 | The speed is average value of cruising cars. |
| Acceleration of cruising car (m/s$^2$) | 0.010 | 0.530 | 0.250 | 0.079 | The acceleration is the average value of cruising cars. |
| Number of accelerations of cruising car (times) | 9.000 | 45.000 | 27.410 | 8.284 | The number of accelerations is defined as cruising cars took acceleration or deceleration times. |
| Number of lane-change of cruising car (times) | 0.000 | 17.000 | 4.790 | 3.582 | The number of lane-change is the times that cruising cars changed lanes. |
| Frequency of lane-change of cruising car | 0.100 | 0.790 | 0.370 | 0.001 | The frequency of lane-change is measured by lane-change times of cruising cars. |
| Distracted time of cruising driver (s) | 0.000 | 19.000 | 3.530 | 22.599 | The time of the driver is distracted by searching for vacated parking lot. |

*3.2. Survey Results from Videotapes*

Through the video observation and calculation, the travel time, volume, and speed of normal traffic flow were obtained. Their average values were 3.01 min, 745.37 veh/h, and 22.13 km/h. The search or cruising time of parked cars was 2 times more likely than travel time of normal traffic flow. The speed of normal traffic was 1.64 times more likely than cruising cars. Cruising for parking has a negative impact on traffic flow.

As shown in Figure 2, comparison between normal vehicle and cruising vehicles was conducted by the time-series analysis. Normal vehicle speed was chosen as a control group. As shown in Figure 2a–c, the shorter the search time by cruising cars, the more frequent speed changes were. The speed of cruising cars was always lower than normal vehicles. As shown in Figure 2b,c, after the search time of cruising cars was more than 12min, the speed change begins to produce periodic oscillations. Cruising for parking would result in a decrease in the overall speed.

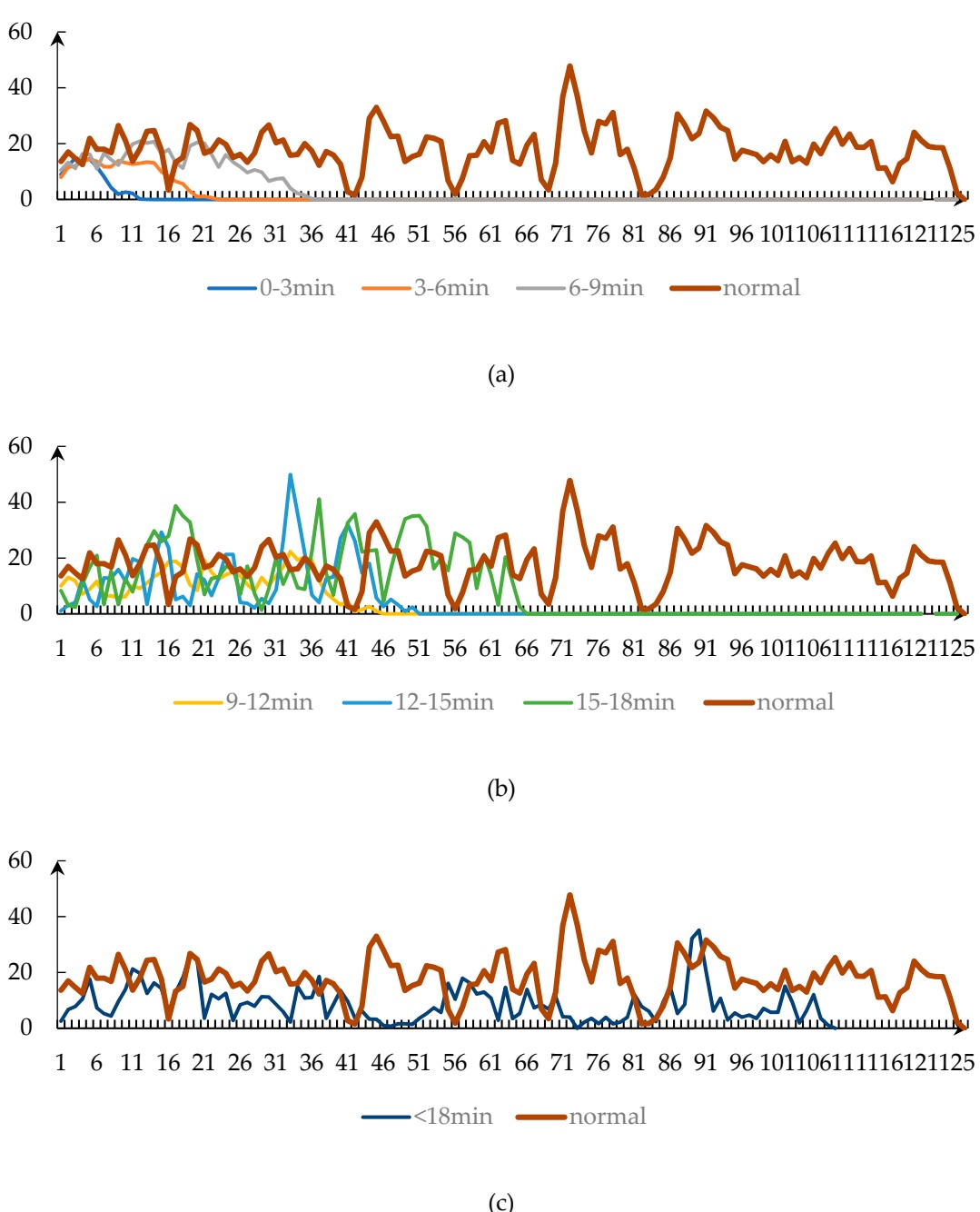

**Figure 2.** Speed of Cruising Car at Different Time Length is Compared with Normal Vehicle Speed. (**a**) Speed of Cruising Car at Short Time Length is Compared with Normal Vehicle Speed; (**b**) Speed of Cruising Car at Long Time Length is Compared with Normal Vehicle Speed; (**c**) Speed of Cruising Car at Over-long Time Length is Compared with Normal Vehicle Speed.

## 4. Methods

The hazard-based duration model has been widely used in biometrics and reliability engineering for decades [22]. The proportional hazard-based duration models represent a type of analytical method that describes the duration of a certain state and how various factors have affected the duration. The duration model has its unique advantages in quantitative analysis of the various factors: (1) Factor analysis. It can not only accurately quantify the role of factors, but also calculate the relative risk of each factor. (2) The single factor influences the utility among different levels. (3) By comparing the relative importance of each factor to the duration, we can get the state law of the duration variable.

(4) Examine the interaction between factors. Because the duration model is the best option to determine the causal relationship of continuous time variables, this paper chooses this method to analyze the influential factors of cruising for parking. Based on the proportional hazard-based duration model, the influence of cruising behavior for parking is analyzed.

*4.1. Hazard-Based Duration Model*

To study duration data, hazard-based models are applied to study the conditional probability of the duration ending at some time, $t$, suppose that the duration has lasted until time $t$. By analogy, let $T$ be a nonnegative random variable representing in the test section, the travel time of the normal cars mixed with cruising vehicles. The cumulative distribution function is shown in Equation (1):

$$F(t) = P(T \le t) = \int_{\infty}^{1} f(u)du \tag{1}$$

where:

$f(t)$ = the probability density function of $T$;

$P$ = probability;

$t$ = some specified time.

Equation (1) gives the probability of passing the section before a certain travel time, $t$.

Let $S(t)$ denote the probability that that the travel duration does not end before $t$, yielding Equation (2):

$$S(t) = P(T > t) = \int_{t}^{\infty} f(u)du \tag{2}$$

As shown in the above equation, $S(t)$ represents the possibility that a vehicle travel time is longer than $t$. It is worth mentioning that $S(t)$ is usually called survivor probability or endurance probability in the duration literature. $S(t)$ is defined as continuous probability in this paper.

In the hazard-based duration model, $T$ is usually expressed as $h(t)$. It is a hazard function, saying that for the instantaneous probability, travel duration in study will be at the end of the infinitesimal time after time t, namely Δt, assuming that the duration hasn't finished before time $t$. The limit definition of $h(t)$ is shown in Equation (3):

$$h(t) = \lim_{\Delta t \to 0} \frac{P(t \le T \le \Delta t | T \ge t)}{\Delta t} \tag{3}$$

The hazard function result is hazard rate or hazard. Specifically, the approximate probability of $h(t)\Delta t$ is the duration terminating in $(t, \Delta t)$, which continues to t. Four distribution functions of $T$ have been mentioned before, including probability density function, cumulative distribution function, continuous probability function, and hazard function. Therefore, the hazard function can be defined by the other three, which is shown in Equation (4):

$$h(t) = \frac{f(t)}{S(t)} = -\frac{d \ln S(t)}{dt} \tag{4}$$

To Integrate Equation (4) from 0 to $t$ and use $S(0) = 1$ to get Equation (5):

$$S(t) = \exp(-\int_{0}^{t} h(u)du) \tag{5}$$

The travel time of normal cars mixed with cruising vehicles is influenced by various factors. The main purpose of this paper accommodates the effects of these influential factors. To define influential factors as a vector of explanatory variables, $x$. Then, introduce the proportional hazard (PH) form,

which specifies the multiplier effects of explanatory variables on some potential (or baseline) hazard function, yielding Equation (6):

$$h(t) = h_0(t)g(x,\beta) \tag{6}$$

where:

$h_0(t)$ = baseline hazard function, which indicates the hazard when not considering the influences of explanatory variables [i.e., $g(x,\beta)$ ];

$g(x,\beta)$ = the known function, which indicates the impacts of explanatory variables;

$\beta$ = an estimable parameter vector of $x$.

In short, it is assumed that covariance influences the baseline hazard to the function $g(x,\beta) = \exp(\beta x)$, proposed by Cox [23]. This specification guarantees the positivity of the hazard function without placing constraints on the signs of the elements of $\beta$. Thus, it is convenient, and the Cox proportional hazard model is shown in Equation (7):

$$h(t) = h_0(t)\exp(\beta_1 x_1 + \beta_2 x_2 + \ldots + \beta_p x_p) \tag{7}$$

*4.2. Model Estimation*

The model of two components $\beta$ and $h_0(t)$ in Equation (7). Cox proposed the partial likelihood method, which is an ingenious way of estimating $\beta$ [19]. Suppose that a random sample consists of k different observed duration data, $t_{(1)} < t_{(2)} < \ldots < t_{(k)}$. Assume $x_{(i)}$ is the variable related to the sample observed at $t_{(i)}$. $R\left[t_{(i)}\right]$ consists of all individuals with duration of at least $t_{(i)}$, denoting the risk set at $t_{(i)}$. Let $l$ denote the serial number of variable $x$. The log-partial likelihood function for estimating $\beta$ is shown in Equation (8):

$$LL(\beta) = \sum_{i=1}^{k}\left\{\beta x_i - \log\left[\sum_{l \in R[t_{(i)}]} \exp(\beta x_l)\right]\right\} \tag{8}$$

The estimation of $h_0(t)$ can refer to Bhat [24], Lee and Wang [25].

The overall goodness of fit of the model estimation is determined by the likelihood ratio (LR) statistics, as shown in Equation (9):

$$x_L = -2\left[L(\beta_0) - L(\hat{\beta})\right] \tag{9}$$

where:

$L(\beta_0)$ = the log likelihood for the null model when all the regression coefficients are set as zero;

$L(\hat{\beta})$ = the log likelihood at convergence with $K$ regression coefficients.

## 5. Empirical Results

*5.1. Factor Selection and Explanation*

The explanatory variables contained in the model are based on field research to obtain visual data on the impact of the traffic performance of cruising for parking on traffic flow. In order to ensure the feasibility of data acquisition, eight explanatory variables were selected in the survey data.

1. Travel time, $T$(min). The travel time reflects traffic conditions under the influence of cruising behavior for parking. Through the survey, the average value of travel time was 3.01 min.
2. Speed, $V$(km/h). The speed is the average speed of traffic on the road section. It reflects the road traffic situation, including cruising vehicles.
3. Volume, $Q$(veh/h). The volume contains cruising vehicles and normal vehicles. It was counted for every road segment corresponding to the average volume.

4. The percent of cruising vehicles, *P* (%). It is defined as the ratio of the number of cruising vehicles to the total number of vehicles. The samples were classified by the categorical variable *P*.

5. Acceleration of cruising car, $a(m/s^2)$. The acceleration of cruising car is obtained from GPS positioning data of cruising vehicles.

6. The number of accelerations of cruising cars, $a_c$(times). The times of acceleration of cruising car were counted for every cruising vehicle acceleration time corresponding to the average number of accelerations of cruising cars.

7. The number of lane-changes of cruising cars, $LC_c$(times). The times of lane-changes of cruising cars were counted for every cruising vehicle change lane time corresponding to the average number of lane-changes of cruising cars.

8. Frequency of lane-change of cruising cars, $LC_F$. It is defined as the ratio of the number of changing lanes of cruising vehicles to the total number of vehicles. The data were recorded by video.

9. Distracted time of cruising driver, *Dt*(s). The time the driver is distracted by searching for vacated parking lot.

The volume, speed, and density are three important parameters in traffic theory. Due to the mixed normal traffic flow with cruising vehicles, the three parameters models are unambiguous. Partial correlation analysis was also used to analyze correlation between travel time and influential factors. There was no collinearity. The relationships between variables were not considered. Moreover, the duration model can consider the interaction between factors. The interaction between the speed, volume, acceleration, and lane-changing of vehicles were also introduced into the duration model. However, that was not significant. Therefore, the speed, volume, acceleration, and lane-changing of vehicles were selected as the independent variables in the proposed model.

*5.2. Estimated Results*

Model estimation used the method of Stepwise regression. Then, the variables *V*, *Q*, *a*, $a_c$, and *Dt* were reserved. Considering the interaction between explanatory variables, the final estimation of the travel time duration model is shown in Table 2. The LR statistic is 377.562 and larger than the Chi-squared statistic, in which situation the overall goodness of fit in the model is proven good. Because the LR statistic has 5 degrees of freedom at any level of significance, it clearly and comprehensively indicates the overall goodness of fit. From the results about statistical significance of each variable, all included variables are statistically significant at the 0.05 level of significance.

**Table 2.** Model Estimation.

| Variable | $\beta$ | Standard Error | Wald Value | Sig. | Exp ($\beta$) | 95%CI for exp ($\beta$) | |
|---|---|---|---|---|---|---|---|
| | | | | | | Lower | Upper |
| *V* | 0.145 | 0.048 | 9.213 | 0.002 | 1.156 | 1.053 | 1.269 |
| *Q* | −0.006 | 0.001 | 40.093 | 0.000 | 0.994 | 0.993 | 0.996 |
| *a* | −5.655 | 0.811 | 48.609 | 0.000 | 0.004 | 0.001 | 0.017 |
| $a_c$ | −0.220 | 0.037 | 34.485 | 0.000 | 0.803 | 0.746 | 0.864 |
| *Dt* | −0.605 | 0.063 | 92.825 | 0.000 | 0.546 | 0.483 | 0.618 |

Figure 3 shows the observed continuance probability, as well as the continuance probability estimated by the model. The curve of the estimated distribution is monotone increasing (i.e., the continuance probability increases with the increase in travel time). Although the overall trend of the observed results is almost the same as that of the estimated, there are also differences between them. Specifically, the observed continuance probability compared with the estimated results is smaller. What's more, the differences between the two distributions can be partly due to the variable effect, which emphasizes the necessity of identifying the continuous probability changes caused by variables.

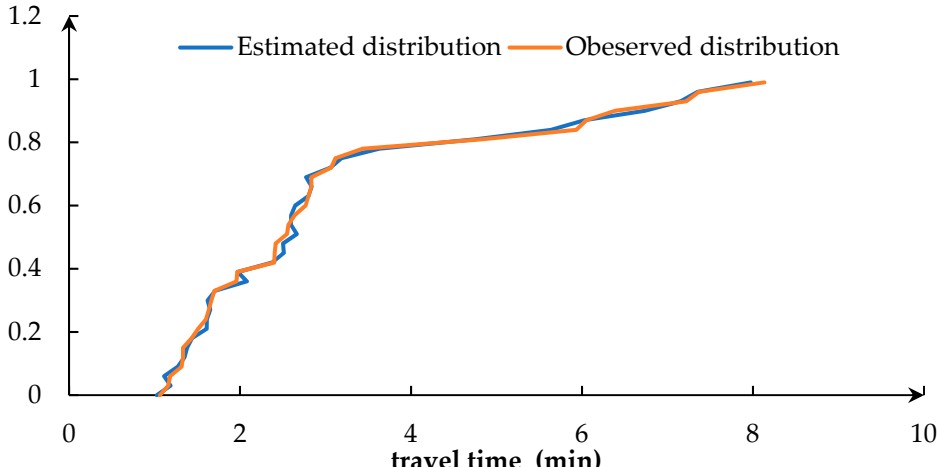

**Figure 3.** Observed and Estimated Continuance Probability Versus Travel Time.

### 5.3. Effects of Explanatory Variables

The explanatory variables effect can be interpreted in a rather straightforward fashion by the signs of the coefficients. If the coefficient is positive, it means that the reduction of the corresponding variable will increase the risk rate, or is equal to a reduction in travel duration. In respect to the magnitude of variable effect, if a variable changes one unit, the hazard changes $[\exp(\beta) - 1] \times 100\%$. As shown in Table 2, the variables $Q$, $a$, $a_c$, and $Dt$ indicated a positive effect on travel time. However, the variable $V$ had a negative impact, indicating that increasing variables could decrease the travel time. In order to evaluate the impacts on which travel duration is affected by explanatory variables, dividing both sides of Equation (7) by $h_0(t)$ can obtain a function of the hazard ratio (HR) (Equation (10)).

$$\frac{h(t)}{h_0(t)} = \exp(\beta x) = \exp(\beta_1 x_{1i} + \beta_2 x_{2i} + \ldots + \beta_n x_{ni}) \tag{10}$$

where:

$x_{ni}$ = the $n$th variable of the $i$th observed vehicle;

$\beta_n$ = the corresponding coefficient.

The HR can represent the multiple relations between the hazard under the variable effects and the hazard when all variables are ignored ($x = 0$).

The variables in the denominator of the left side of Equation (10) are standardized about the mean and yield Equation (11).

$$\frac{h(t)}{h(t)'} = \exp[\beta_1(x_{1i} - \overline{x_1}) + \beta_2(x_{2i} - \overline{x_2}) + \ldots + \beta_n(x_{ni} - \overline{x_n})] \tag{11}$$

where $h(t)'$ = hazard with the average variables;

$\overline{x_n}$ = average of the $n$th variable for the samples.

Equation (11) is the relative hazard ratio (RHR), or called the relative hazard index.

In order to quantitatively analyze the effects of cruising vehicles, it can be presumed that a variable is in the positive or negative state, while other variables are taken as the average. Then, calculate the HR or RHR for each variable. In addition, the observed vehicles are in a positive or negative state compared with the usual state can describe the multiples of the hazard by the RHR. The HR can be used to describe the multiples of the hazard when the observed vehicles are in a favorable condition (travel time can be reduced) compared with the unfavorable condition (travel time can be increased). The influence of the traffic on the travel time can be analyzed quantitatively based on the RHRs and

HRs in specific conditions. Table 3 shows the specific conditions and corresponding HRs and RHRs. Figure 4 shows the RHRs for five variables ($V$, $Q$, $a$, $a_c$, and $Dt$) to visualize their effects.

**Table 3.** Analysis of Variables Related to Cruising for Parking Characteristics.

| Variable | Mean | Variable Value | | Relative Hazard Ratio | | Hazard Ratio |
|---|---|---|---|---|---|---|
| | | Unfavorable | Favorable | Low Hazard | High Hazard | |
| $V$ | 22.13 | 6.48 | 49.87 | 0.10 | 55.74 | 539.67 |
| $Q$ | 757.42 | 162.78 | 1800.00 | 0.01 | 35.44 | 77.95 |
| $a$ | 0.25 | 0.01 | 0.53 | 0.20 | 3.98 | 6.46 |
| $a_c$ | 27.41 | 9.00 | 45.00 | 0.02 | 57.43 | 126.47 |
| $Dt$ | 3.52 | 0.00 | 19.00 | 0.00 | 8.46 | 98,223.42 |

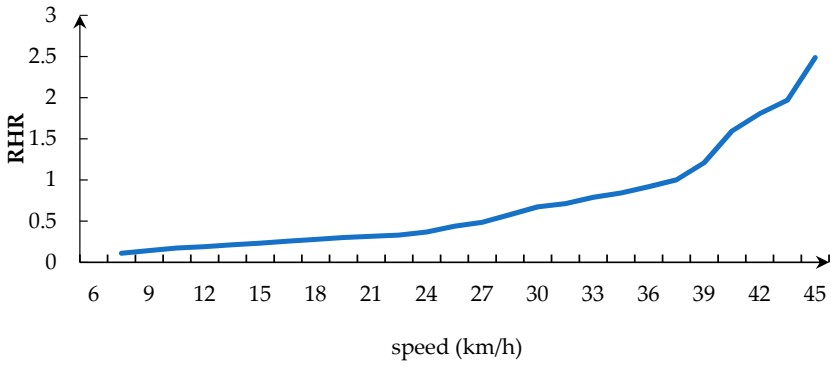

(a)

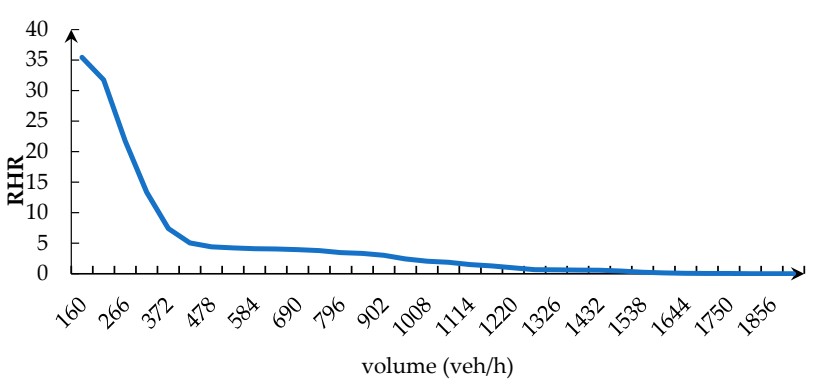

(b)

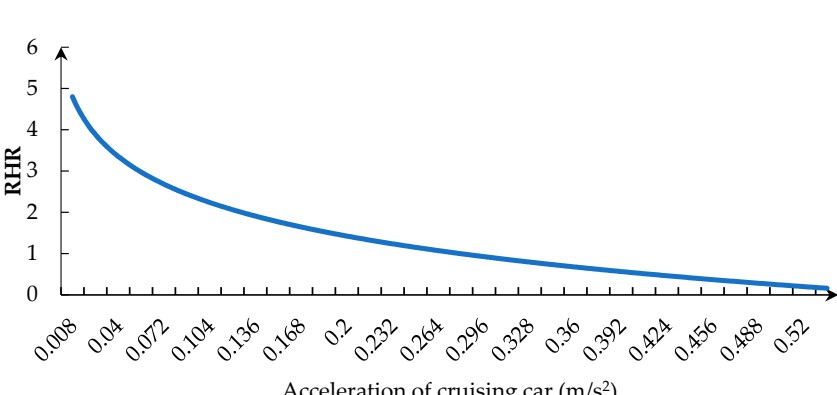

(c)

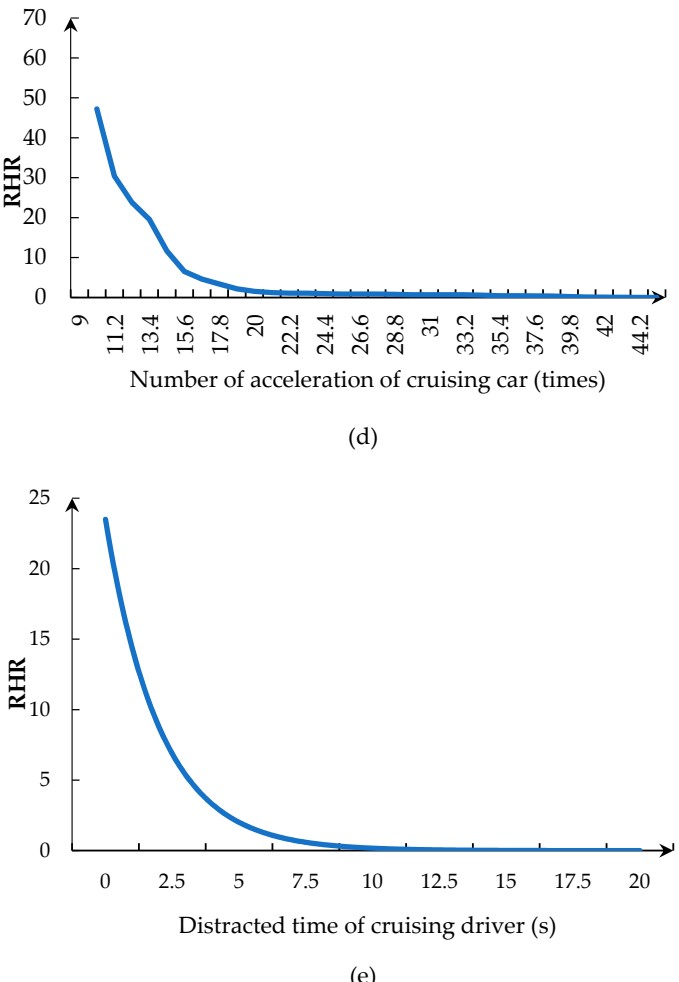

(d)

(e)

**Figure 4.** Variables Related to RHR. (**a**) The relationship between speed and RHR; (**b**) The relationship between volume and RHR; (**c**) The relationship between acceleration of cruising car and RHR; (**d**) The relationship between number of cruising car and RHR; (**e**) The relationship between distracted time of cruising car and RHR.

### 5.3.1. Effect of Speed

The effect of the speed (*V*) indicated that the increasing effective speed can increase the hazard or decrease the continuance probability (shown in Figure 4a). The RHR is lower than 0.5 when the cruising vehicle is under 22.13 km/h. In other words, the slow growth of the curve indicates no significant impact when the cruising vehicle is under 22.13 km/h. The influence of speed on shortening travel time becomes obvious when the cruising vehicle is higher than 22.13 km/h. The travel time is affected by speed and factors besides. When the vehicle speed is above 22.13 km/h, it becomes the main influence.

### 5.3.2. Effect of Volume

The effect of the volume (*Q*) indicated that the increasing effective volume can decrease the hazard or increase the continuance probability (shown in Figure 4b). The RHR is lower than 5 when the volume is under 425 veh/h. The influence of volume on shortening travel time becomes step-down when the volume is higher than 425 veh/h. Through the analysis of traffic flow three-parameter relationship, the road traffic began to jam after the volume reaches a particular value. Therefore, the RHR is low and gentle by this situation.

### 5.3.3. Effect of Acceleration of Cruising Car and Number of Accelerations of Cruising Cars

The effect of the acceleration of cruising cars ($a_c$) and number of accelerations of cruising cars indicated that the increasing effective acceleration of cruising cars and number of accelerations of cruising cars can decrease the hazard or increase the continuance probability (shown in Figure 4c,d). In particular, the acceleration of cruising cars differed from normal cars. Due to searching the park space during driving, the driver of the cruising car was distracted, and often changed the acceleration subjectively. Normal cars were affected by the frequency of acceleration-change behavior of cruising cars. Travel time of traffic flow with mixed normal cars and cruising cars increased. In general, search time includes acceleration from the beginning to the end. The higher the acceleration, the shorter the acceleration time. Search time increases with a greater number of accelerations of cruising cars. Search time is an important reason for the increase in travel time. The average cruising time is 1.52min when the number of accelerations of cruising cars is less than 20 times. The average cruising time is 3.64min when the number of accelerations of cruising cars is more than 20 times. In Table 3, results show that a higher ratio of acceleration of cruising cars and number of accelerations of cruising cars would increase travel time (RHR is 6.46 and 126.47).

### 5.3.4. Effect of Distracted Time of Cruising Driver

The effect of the number of accelerations of cruising cars and distracted time of cruising drivers ($Dt$) indicated that the increasing effective volume can decrease the hazard or increase the continuance probability (shown in Figure 4e). The RHR is 98,223.42 for the particular conditions. The probability of cruising vehicles timing 19s absent-minded time is 98,223.42 times the probability of these vehicles timing not distracted time of cruising driver. A long distracted time of a cruising driver increases traffic congestion and traffic accidents.

In this paper, multiple regression model was used to compare with the Cox proportional hazard-based duration model. The multiple regression model is Equation (12).

$$Y = \beta_0 + \beta_1 x_1 + \beta_2 x_2 + \cdots + \beta_n x_n + \varepsilon \tag{12}$$

In this paper, $Y$ is travel time as the dependent variable, and other variables are independent variables. The coefficients $\beta$ were calibrated in Table 4.

**Table 4.** Parameter Calibration Results of Vehicle Velocity Regression Mode.

| Variable | $\beta$ | Sig. |
|----------|---------|------|
| Constant | −2.185 | 0.000 |
| $V$ | 0.057 | 0.000 |
| $Q$ | −0.000135 | 0.013 |
| $a$ | 1.359 | 0.000 |
| $P$ | 5.338 | 0.000 |
| $a_c$ | 0.058 | 0.000 |
| $CL_F$ | 1.512 | 0.010 |
| $Dt$ | 0.144 | 0.000 |

The analysis results of multiple regression model and the Cox proportional hazard-based duration model were compared in Figure 5. The average value of the observed travel time was 3.02 min. The average travel time of the Cox proportional hazard-based duration model was 3.02 min. The average travel time of the multiple regression model was 3.19min. Meanwhile, the average error of the Cox proportional hazard-based duration model and observed value was 0.44%. The average error of the multiple regression model and observed value was 8.5%. Obviously, the error of the former was smaller. In general, the Cox proportional hazard-based duration model worked better as shown in the Figure 5.

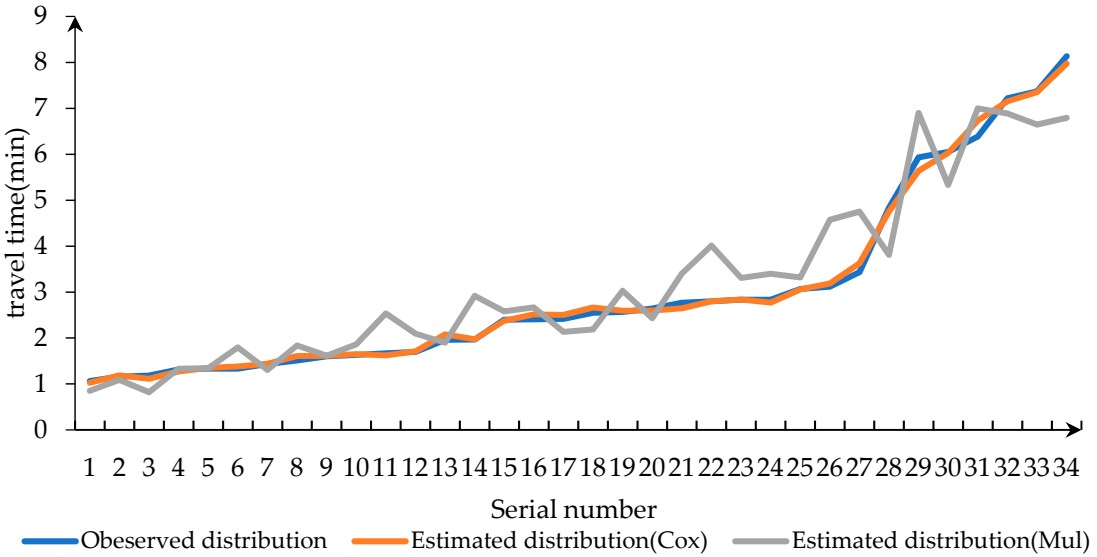

**Figure 5.** Comparison of Model Results.

*5.4. Model Application*

The estimated model can be applied to estimate the distribution of travel time under different groups. This paper grouped data by cruising vehicle proportions. If the cruising vehicle proportions ranges from 0 to 50%, the new distribution of the continuance probability would be as shown in Figure 6. The differences between the curves showed a significant impact on the travel time, which depended on the traffic performance of cruising for parking. The cruising vehicle proportions used in this study varied from 0% to 50% (0%, 5%, 10%, 15%, 20%, 25%, 30%, 40%, and 50%), the corresponding groups were 0, 1, 2, 3, 4, 5, 6, and 7. Travel time would decrease by 85.63% if the cruising vehicle proportions decreased from 50% to 0%. Similarly, the hazard-based duration methodology can obtain the influence of impact factors. Travel duration adjustment was based on the above effects. Therefore, the operational analysis improving benefits the hazard-based duration model. For instance, the estimated travel time could be used as an index to evaluate the influence of the traffic performance of cruising for parking, and a firm basis for improving the traffic congestion. Before applying, the model should use the specified field data to estimate. In addition, the explanatory variables should be chosen flexibly according to the research aim and the actual traffic situation.

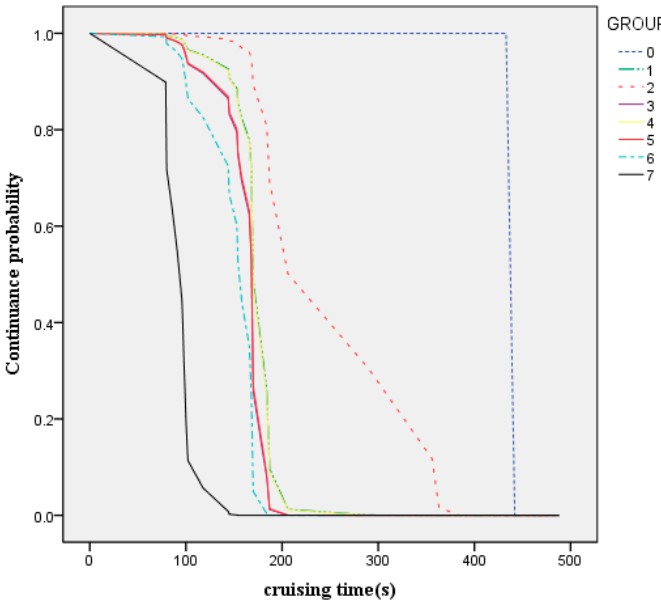

**Figure 6.** Variables Related to Hazard Ratios.

## 6. Discussion

Scholars began to study the traffic performance of cruising for parking when old photographs and postcards show cars parked bumper to bumper on busy streets. These researches are mainly carried out from three aspects: The proportion of parking cruising vehicles in the traffic flow, the proportion of parking space occupied by a certain value, the relationship between parking cruising behaviors and parking fee prices, etc., and preliminary progress has been made, laying a foundation for the following researches. However, there is still a lack of research on the characteristics of cruising vehicles, such as search time, speed, lane change frequency, etc., which is worth exploring for seeking more accurate methods for identifying cruising vehicles. This paper studied the traffic performance of cruising for parking mechanism from a microscopic perspective. It was in order to propose implementation strategies.

From the hazard-based duration methodology's results, the influential factors on the intention of travel time were the speed, volume, acceleration, and the number of accelerations of cruising cars, and distracted time of cruising drivers. In particular, the speed of cruising cars differed from normal cars. Due to searching the parking space during driving, the driver of a cruising car was distracted and often kept driving at low speed subjectively. Therefore, the speed was the one of causes of travel time. Speed had the most direct effect on travel time. The longer the travel time, the worse the traffic. The following measures are proposed to reduce travel time.

Firstly, making parking space more visible to reduce distracted time of cruising drivers. For example, eye-catching lines and reflective signs of parking lots could be set up for high visibility. A distance-based voice reminder of parking lot information should be equipped in the parking guidance system. When the driver searches for a parking space, there will be a time of distraction. The faster the driver finds the parking space, the less time it takes to travel.

Secondly, parking lot entrances should not be set up on the main traffic road. When the driver encounters the entrance of the parking lot, he will slow down to see if there is any parking space. If there is no parking space, the driver accelerates to leave. At the same time, the parking facilities should be placed on the branch road to disperse the pressure on the arterial road. This will reduce the number of vehicle acceleration and low-speed travel time so as to achieve the purpose of reducing the influence of the main road.

Finally, accelerate the construction of intelligent parking navigation system. Drivers can get parking space as early as possible to reduce search time. Reduced time and acceleration times when

the cruising vehicle has a clear destination. And ultimately achieve the purpose of reducing traffic so as to reduce traffic congestion.

## 7. Conclusions

This paper studied travel time of traffic flow under the impact of cruising for parking based on a hazard-based duration model. Cruising vehicle survey was undertaken in Tianyi square CBD test sections. Meanwhile, the traffic performance of cruising for parking and effects of cruising for parking were recorded. The methodology used a framework of proportional hazard. It allowed different individuals to have various travel time according to traffic conditions and the traffic performance of cruising for parking. Above all, the hazard-based duration methodology obtained the influence of impact factors associated with the traffic performance of cruising for parking and traffic conditions. For this reason, it can express how the cruising for parking affects the distribution of travel time. Through the model verification provided in this paper, it shown that the hazard duration method was appropriate for the impact analysis of cruising for parking.

In this paper, the following views are obtained through the study of the decisive factors that the distribution of travel time under the influence of cruising for parking. First of all, the research evidence proves that various related factors determine the effect of cruising for parking. The distribution of travel time reflects this influence. Any change factors could affect the travelling time. Secondly, the speed shows a positive effect on the travel time; cruising volume, acceleration of cruising cars, number of accelerations of cruising cars, and distracted time of cruising drivers. However, it has shown a negative effect on travel speed. The speed and number of accelerations of cruising cars are the most significant influencing factors. Finally, the hazard-based duration model is well suitable for the study of cruising for parking. The distribution of travel time can quantitatively represent the influence of cruising for parking. Therefore, the model can be used for quantitative analysis of the influence of cruising for parking by the distribution of travel time estimated.

In the future, the relationship between traffic congestion and cruising for parking is worth studying. At the same time, studying the relationship needs consider autonomous vehicle environments in the future [26–28]. Consider using cellular automaton model to further refine the influence of cruising for parking [29]. These theories can to deepen the awareness of cruising for parking, and we hope to have some help in the design of parking facilities and parking planning.

**Author Contributions:** The authors confirm contribution to the paper as follows: Formulation or evolution of overarching research goals and aims: Y.Z., X.Y. (Xiaofei Ye); data curation and investigation: X.Y. (Xingchen Yan), T.W. (Tao Wang); formal analysis and methodology: Y.Z., X.Y. (Xiaofei Ye), J.C. (Jun Chen); writing—original draft: Y.Z., X.Y. (Xiaofei Ye). All authors have read and agreed to the published version of the manuscript.

**Funding:** Natural Science Foundation of Zhejiang Province, China Grant number (No.LY20E080011), Natural Science Foundation of China (No.71971059, 71701108 & 71861006), National Key Research and Development Program of China—Traffic Modeling, Surveillance, and Control with Connected & Automated Vehicles, China Grant number (No.2017YFE9134700), Natural Science Foundation of Ningbo, Zhejiang Province, China Grant number (No.2017A610139), Basic Research Program of Science Grant number (No.BK20180775), and Technology Commission Foundation of Jiangsu Province, China Grant number (No.BK20170932).

**Acknowledgments:** The authors thank the relevant institutions for funding this project. The authors thank drivers for providing this work data.

**Conflicts of Interest:** The authors declare no conflict of interest.

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
