# Peer review of "Impact of Cruising for Parking on Travel Time of Traffic Flow"

_sustainability, doi:10.3390/su12083079_

Round 1

Reviewer 1 Report

1. Authors identified the influential factors on the intention of travel time. Data collection process was correct but the experiment description shall be more detailed. For example, all out of 14 days were working days?

2. A strong point of the discussion part are suggestions for the improvement of parking facilities to reduce travel time.

3. Theoretical parking modeling is rare (including question of cruising) in the literature and there is a minor evidence on this very important topic. Nevertheless,  Authors should improve the literature review, using i.e. following papers:

Zhen (Sean) Qian, Ram Rajagopal: Optimal dynamic parking pricing for morning commute considering expected cruising time. "Transportation Research Part C"  48 (2014),

Jos van Ommeren, Derk Wentink, Jasper Dekkers: The real price of parking policy. "Journal of Urban Economics"  70 (2011),

Xiang Yan, Jonathan Levine, Robert Marans: The effectiveness of parking policies to reduce parking demand pressure and car use. "Transport Policy" 73 (2019),

Richard Arnott, John Rowse: Downtown parking in auto city. "Regional Science and Urban Economics"  39 (2009),

N. Geroliminis: Cruising-for-parking in congested cities with an MFD representation. "Economics of Transportation" Volume 4, Issue 3, September 2015,

E. Inci: A review of the economics of parking. "Economics of Transportation" 4 (2015),

O. Cats, Ch. Zhang, A. Nissan: Survey methodology for measuring parking occupancy: Impacts of an on-street parking pricing scheme in an urban center. "Transport Policy"  47 (2016).

Very minor remarks:

Line 139. space needed between words "3.53s." and "More"

Line 366. Space needed between words "0%." and "Similarly"

Author Response

Thank you very much for your comments and suggestions.

Reviewer 2 Report

This paper studies the impact of cruising for parking on travel time based on the hazard-based duration method with the GPS and camera data. For the method, there is no contribution. It is simply an application of hazard-based duration model in the literature. Other comments are follows.

- The observation is that "speed is the most direct effect on travelling time". It is correct (even without the need of data) but not interesting. Does the speed is the cause (drivers actively change the speed without impact of ahead traffic) or an effect (drivers change or reduce speed due to congestion ahead) of travel time? 

- The variables are related to speed, volume, acceleration and lane-changing of vehicles. Should we choose independent variables? In traffic theory, there are relation between speed and volume. Volume and the number of lanes are also relevant. 

- Line 306: what is the definition of "favorable condition"?  

- Figure 5 is unclear about what is compared. What are the meaning of x-axis and y-axis?

- Line 63: "The findings could explain how cruising cars adds to the normal traffic that is already congested and quantify the worst situation of the influence". Please elaborate more about the "how" as it is unclear from the writing.

- Line 116: I am unclear about the citation [17] there. Is this data is from this work? How is this work related to this paper?

- Line 163: "the duration model is the best option to determine the causal relationship of continuous time variables". Please elaborate more about this point, e.g., why it is the best option, what is its advantage compared to other methods?

Author Response

Thank you for you help!

Reviewer 3 Report

The manuscript discussed parking management. The parking management can be correlated with an environmental and societal issues as part of sustainability. The paper presented the factors of driving behavior in order to find a parking spot.

The following points should be addressed:

  1. In Introduction, the benefits of the study and the relevance to the sustainability should be stated.
  2. In Literature Review, what are the gaps between the study and previous studies to fill? 
  3. The authors tested more than 450 in Line 125 on Page 3. The number should be a fixed one, not more than or fewer than 450 since that is a sample size.
  4. In Table 1, if the sample size n=450 and have an average value, why don't you provide variability of the survey data?
  5. Figure 2 is very busy, can you split it into 2-3 graphs?
  6. in Line 285 on Page 8, V is only variable which has a negative impact. so "other variables" should be "another variable"?
  7. In Equation (11), it will be better to have "/" on the LHS to differentiate from the average of h(t).
  8. In line 299 on Page 9, mean(xn) = average of the set i for nth sample?
  9. In Line 328-330 on Page 10, it can also be: Decrease of the hazard can increase the number of accelaration of cruising car. Be clarify the cause and effect!
  10. On Figure 5, provide the titles of x and y-axis.
  11. Under References, what do the stars mean for References 22-24?
  12. Line 248 on Page 7. the sentence is not complete. 

Author Response

(The authors gave the same response as above.)

Round 2

Reviewer 2 Report

The response to my comments gives more clarity. 

There are still some typos and grammatical issues in the paper.

Other than that, I have no more comments.

Author Response

Dear Editors and Reviewers:

  Thank you for your help!

This manuscript is a resubmission of an earlier submission. The following is a list of the peer review reports and author responses from that submission.